# Second-generation lung-on-a-chip with an array of stretchable alveoli made with a biological membrane

Pauline Zamprogno[1], Simon Wüthrich[1], Sven Achenbach [1], Giuditta Thoma[1], Janick D. Stucki[1,2], Nina Hobi[1,2], Nicole Schneider-Daum[3], Claus-Michael Lehr [3], Hanno Huwer[4], Thomas Geiser[5], Ralph A. Schmid[6] & Olivier T. Guenat [1,5,6✉]

The air-blood barrier with its complex architecture and dynamic environment is difficult to mimic in vitro. Lung-on-a-chips enable mimicking the breathing movements using a thin, stretchable PDMS membrane. However, they fail to reproduce the characteristic alveoli network as well as the biochemical and physical properties of the alveolar basal membrane. Here, we present a lung-on-a-chip, based on a biological, stretchable and biodegradable membrane made of collagen and elastin, that emulates an array of tiny alveoli with in vivo-like dimensions. This membrane outperforms PDMS in many ways: it does not absorb rhodamine-B, is biodegradable, is created by a simple method, and can easily be tuned to modify its thickness, composition and stiffness. The air-blood barrier is reconstituted using primary lung alveolar epithelial cells from patients and primary lung endothelial cells. Typical alveolar epithelial cell markers are expressed, while the barrier properties are preserved for up to 3 weeks.

[1] Organs-on-Chip Technologies Laboratory, ARTORG Center, University of Bern, Bern, Switzerland. [2] AlveoliX AG, Bern, Switzerland. [3] Drug Delivery (DDEL), Helmholtz-Institute for Pharmaceutical Research Saarland (HIPS), Saarbrücken, Germany. [4] SHG Clinics, Department of Cardiothoracic Surgery, Völklingen Heart Center, Völklingen, Germany. [5] Department of Pulmonary Medicine, University Hospital of Bern, Bern, Switzerland. [6] Department of General Thoracic Surgery, University Hospital of Bern, Bern, Switzerland. ✉email: olivier.guenat@artorg.unibe.ch

Organs-on-chips (OOCs) are emerging as predictive tissue modelling tools and as a credible alternative to animal testing. These micro-engineered cell-based systems provide cells with an environment that closely resembles their native in vivo milieu[1–3]. Tissue models of physiologically healthy or pathological primary cells from patients have been established, and are robust enough to permit applications such as drug screening[4–6]. Micro-engineered systems with an integrated membrane in a microfluidic setting have been reported to model various barrier tissue interfaces, such as those of the lung alveoli, the brain and the gut[7]. By implementing a flexible polymeric membrane in such microfluidic systems, mechanical forces, such as those induced by breathing, could be reproduced[8–10].

Although these systems represent a crucial advance in cell culture research, they are still far from mimicking the whole in vivo intricacy. An important in vivo feature that is not reproduced in air–blood barrier models is the array of tiny alveoli. Indeed, reported lung-on-a-chip from the first generation[8,9] simulate a single alveolus with an epithelium area much larger than that of an alveolus in vivo. Given the close relationship between lung microstructure, mechanical forces, alveolar epithelial phenotype and lung functions[11,12], the emulation of the alveolar network would be of great benefit. A further limitation of those systems is the use of an artificial basal membrane made of a thin, porous and stretchable polydimethylsiloxane (PDMS) film. Although PDMS has good elastic, optical and biocompatible properties, it can distort the biochemical microenvironment through high adsorption and absorption levels of small molecules[13,14]. In addition, PDMS differs in important ways from the molecular composition and intrinsic stiffness of the native extracellular matrix (ECM), which is known to affect cellular phenotype and homoeostasis[15,16]. The complex ECM environment provides the structural basis for cellular growth, and influences cellular morphology, functionality, differentiation and other traits[17,18]. The role of the ECM in tissue development and function is closely associated with its composition and properties[19]. The replacement of PDMS as culturing membrane with a material made of ECM molecules would therefore be a significant step towards emulating in vivo-like tissue barriers and functions.

Hydrogels receive a strong interest to recreate the chemical composition and structure of the native ECM in cell culture systems[20,21]. Their intrinsic properties, including mechanical features, chemical composition and porosity, make them ideal candidates to supersede PDMS membranes[22]. However, the creation of thin membranes made of ECM molecules, with stretchable properties to mimic the cyclic mechanical strain of the lung alveolar barrier, is technically challenging. Dunphy et al.[23] developed a stretchable, soft collagen–elastin (CE) membrane for tissue engineering. However, with a thickness of about 1 mm, it was developed to evaluate the mechanical properties of the material and not to mimic the air–blood barrier. More recently, collagen vitrified membrane[24] has been integrated into microfluidic devices for use as cell culture substrates[25,26], but, to the best of our knowledge, no stretchable membranes have been reported so far.

Here, we report about a lung-on-a-chip of the second generation that mimics the following central aspects of the air–blood barrier: (1) an array of alveoli, with dimensions similar to those found in vivo and (2) a biological membrane made of proteins of the lung ECM, collagen and elastin, enabling the membrane to be (3) biodegradable and (4) stretchable. The creation of the biological membrane is based on a bottom-up approach fabrication technique. The membrane is formed by drop-casting a CE solution onto a gold mesh, where it spreads and is maintained by surface tension. The thin gold mesh, with a pore size of 260 µm, is used as the scaffold supporting the array of 40 alveoli. The resulting membrane is stable and can be cultured on both sides

for weeks. Its permeability further allows cells to be cultured at the air–liquid interface, and its elastic properties mimic the respiratory motions by mechanically stretching the cells. Results with primary human alveolar epithelial cells from patients cocultured with primary human lung endothelial cells demonstrate that the air–blood barrier functions can be maintained and used experimentally in a resilient and reproducible manner. This proto-physiological membrane opens the way to new lung-on-a-chip and OOC devices that enable the mimicry of biological barriers with a new level of analogy to whole organ systems.

## Results

**Production of a thin, biological and stretchable membrane.** A simple process was used to create the thin biological membrane (Fig. 1). A drop of CE solution was pipetted onto a 2-mm diameter and 18-µm-thin gold mesh (Figs. 1c and 2a) made of an array of 40 regular hexagons, with sides of 130 µm separated by 30-µm-wide walls. Once pipetted onto the mesh, the CE drop was maintained on its top by surface tension forces. After a gelation step at 37 °C, the CE solution dries out at room temperature within 2 days. While water evaporates from the drop, surface tension forces and residual forces counteract gravity force

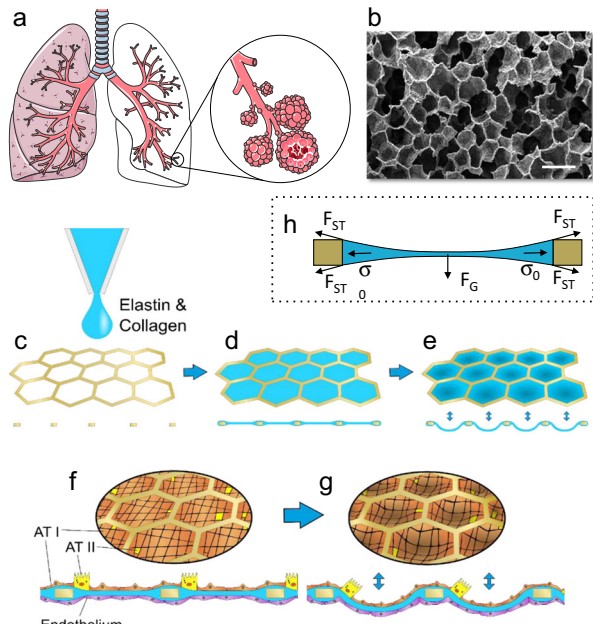

**Fig. 1 Second-generation lung-on-a-chip: creation of the lung alveoli array. a** Schematic of the respiratory tree-like structure ending with alveolar sacs (adapted from https://smart.servier.com/smart_image/lungs-7/, https://smart.servier.com/smart_image/lungs-11/ and https://smart.servier.com/smart_image/lungs/; Servier Medical Art by Servier; https://creativecommons.org/licenses/by/3.0/). **b** SEM picture of a slice of human lung parenchyma with tiny lung alveoli and their ultrathin air–blood barrier (courtesy of Prof. Dr. Peter Gehr, Institute of Anatomy, University of Bern; scale bar: 500 µm). **c**, **d** Schematic of the production of the CE membrane used in the second generation lung-on-a-chip. A thin gold mesh with an array of hexagonal pores of about 260 µm is used as a scaffold, on which a drop of collagen–elastin solution is pipetted. **e–g** The collagen–elastin gel forms a suspended thin membrane that can be stretched at the alveolar level by applying a negative pressure on the basolateral side of the membrane. **f**, **g** Type I (ATI) and type II (ATII) primary human lung alveolar epithelial cells are cocultured with lung endothelial cells on the thin collagen–elastin membrane. **h** Schematic of the force balance during the drying of the membrane. $F_{ST}$, $F_G$ and $\sigma_o$ stand for surface tension force, gravity and residual stress, respectively.

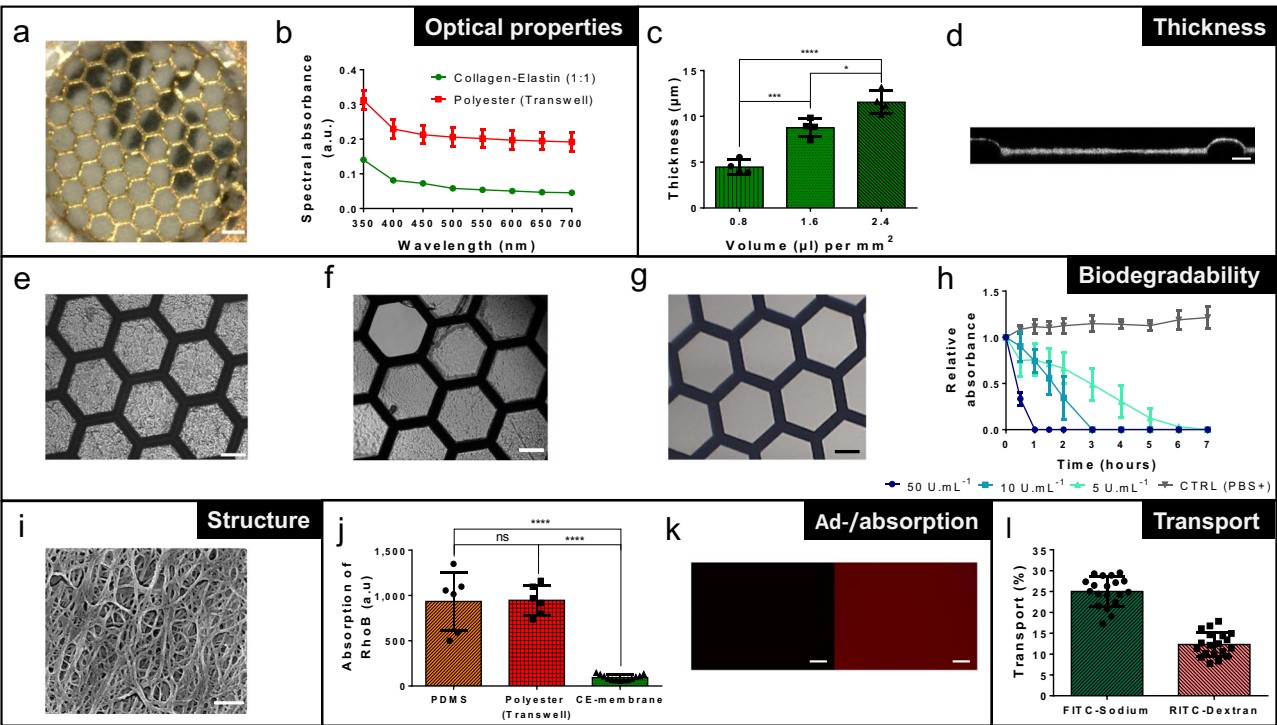

**Fig. 2 Properties of the thin biological membrane. a** Optical clarity of a 10-µm-thin CE membrane integrated in the gold mesh. Scale bar: 200 µm. **b** Comparison of the spectral absorbance of the CE membrane ($n = 4$) and of a polyester membrane (Transwell insert 0.4 µm pores sizes) ($n = 3$). **c** Characterization of the CE-membrane thickness in function of the collagen–elastin solution volume pipetted on top of the gold mesh ($n = 4$). **d** Cross-section of the CE membrane visualized via confocal microscopy. Scale bar: 20 µm. **e** Picture of an array of several hexagons with a CE membrane. Scale bar: 100 µm. **f** Local disruption (top left hexagon) of a membrane after being exposed to 10 U mL$^{-1}$ MMP-8 in PBS+ during 1 h and stretched at −2 kPa. Scale bar: 100 µm. **g** Totally disrupted membrane after being exposed during 1 h to 50 U mL$^{-1}$ of MMP-8 in PBS+. Scale bar: 100 µm. **h** CE-membrane degradation in function of the time and of the MMP-8 concentrations at 550 nm ($n = 4$). **i** SEM picture of the collagen and elastin fibers of the CE membrane. Scale bar: 500 nm. **j** Difference of rhodamine B (10 µM) absorption between a 10-µm-thin CE membrane ($n = 13$), a 10-µm-thin PDMS membrane ($n = 6$) and a polyester porous membrane (Transwell insert, 0.4 µm pores sizes) ($n = 6$). **k** Pictures of CE membrane (left) and a PDMS membrane (right) after being exposed to RhoB for 2 h. Scale bar: 200 µm. **l** Transport of FITC–sodium and RITC–dextran molecules across the CE membrane ($n = 19$).

enabling the suspended membrane to form (Fig. 1h). During this step, the collagen molecules self-assembled into fibrils which, as the natural ECM, provide a structural support (Fig. 2i). The fibrils had an average size of 45.7 ± 18.9 nm (Supplementary Fig. 1). Figure 2d, e illustrates the dried CE membrane with a thickness of only a few micrometres that is suspended on the hexagons array. Once dried, the membrane was integrated into a microfluidic chip, where it was sandwiched between two microfluidic parts, a top part in PDMS with an apical reservoir and a bottom part in polycarbonate that formed the basolateral chamber (Supplementary Fig. 2). The dried membranes are robust and can be stored for at least 3 weeks at room temperature. The membranes are rehydrated by submersion in cell culture medium for 2 h prior to cell seeding.

**Properties of the CE membrane.** The thickness of the membrane was evaluated using reflective light. With a CE ratio of 1:1, the thinnest membrane obtained had a thickness of 4.5 ± 0.8 µm for a pipetted CE solution volume of 0.8 µL mm$^{-2}$ (Fig. 2c). When the pipetted volume was doubled (1.6 µL mm$^{-2}$), the thickness of the membrane also doubled (8.8 ± 1 µm). A thickness of 11.5 ± 1.2 µm was obtained with 2.4 µL mm$^{-2}$. Decreasing the elastin concentration (2:1 ratio) resulted in a reduction of the membrane thickness (Supplementary Fig. 3), to the detriment of its viscoelastic properties (Fig. 3c). The membrane thickness was homogeneous within each hexagon. Variation in membrane thickness across the array was less than 20% (Supplementary Fig. 4) with a

pipetted volume of 1.6 µL mm$^{-2}$. Confocal imaging (Fig. 2d) confirmed these findings.

The optical properties of the CE membrane were assessed by light spectrometry. The CE membrane performed better than a polyester (PET) membrane of standard Transwell inserts. The 10-µm-thin CE membrane absorbed about 10% of visible light, whereas a 10-µm-thin PET membrane with 0.4 µm pores used in inserts absorbed about 20% (Fig. 2b). This low absorbance level was also obtained for a 2:1 ratio CE membrane and for a collagen membrane (Supplementary Fig. 5). The excellent optical properties of the CE membrane were qualitatively confirmed by text placed at the backside of the membrane that was easily read from the apical side (Fig. 2a).

The biodegradability of the membrane was evaluated by optical microscopy. Upon exposure to all the tested concentrations of MMP-8, a matrix metalloproteinase secreted by neutrophils, the CE membrane was totally degraded with no remaining CE leftovers (Fig. 2e–g). The degradation time was found to be inversely proportional to the MMP-8 concentration (Fig. 2h). At 50 U mL$^{-1}$, the membrane was degraded in less than 1 h, while more than 6 h were required to degrade the membrane with MMP-8 at 5 U mL$^{-1}$.

Absorption and adsorption of small molecules on the membrane were tested using exposure to rhodamine B, a molecule that has often been used to show the limitation of PDMS[27,28]. Compared with PDMS and with the PET membranes of similar thicknesses, the CE membrane absorbed much less

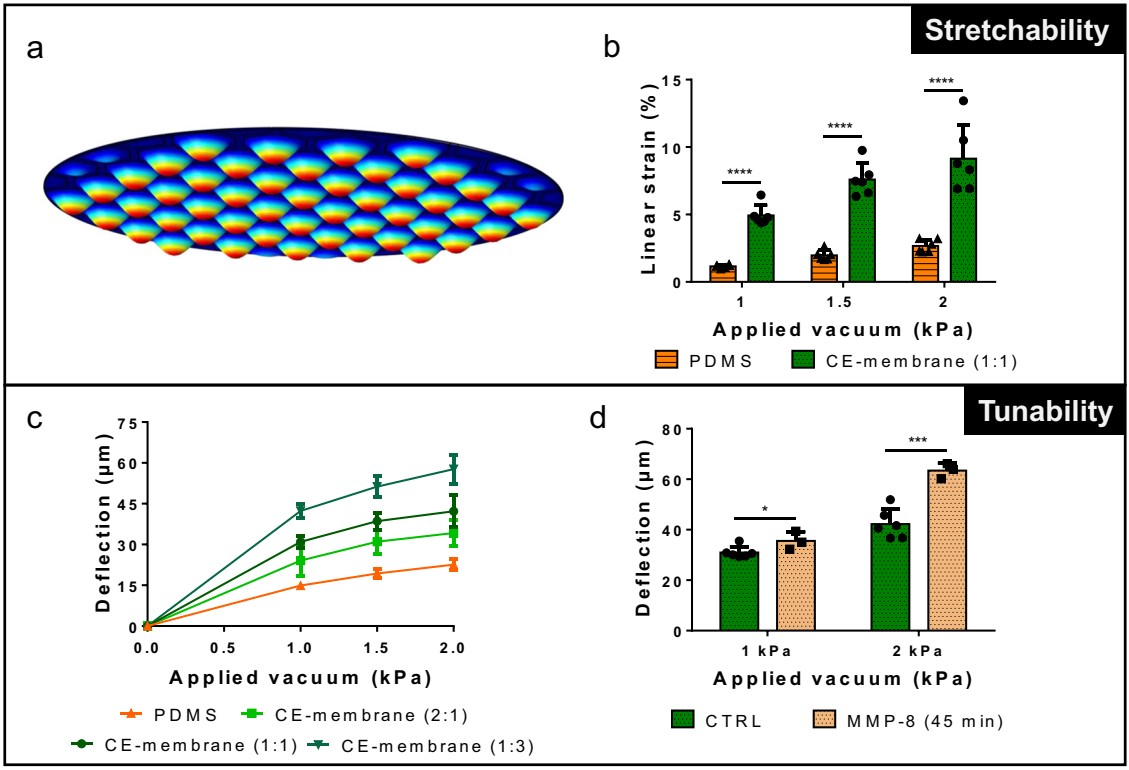

**Fig. 3 Membrane flexibility. a** Numerical simulation of the deflection of the CE-membrane array. **b** Linear strain inside the CE membrane in function of the applied vacuum ($n = 6$). A 10-μm-thin PDMS membrane was taken as reference ($n = 6$). **c** Deflection of CE membranes of various compositions in function of an applied vacuum ($n = 6$ for CE membrane (1:1) and CE membrane (2:1) and $n = 4$ for CE membrane (1:3)). A 10-μm-thin PDMS membrane was taken as reference ($n = 6$). **d** Deflection of CE membrane exposed to MMP-8 in function of an applied vacuum ($n = 6$ for CE membrane (1:1) (CTRL) and $n = 3$ for treated membranes (MMP-8 (45 min))).

rhodamine B. After 2 h of immersion in 10 μM rhodamine B, the number of fluorescent molecules ab-/adsorbed was about 90% lower in a 10-μm-thin CE membrane than in the PDMS and the PET membranes ($p$ value < 0.0001) (Fig. 2j). The absorptions/ adsorptions of all polymeric membranes tested are higher than those of all biological membranes (Supplementary Fig. 6). This absorption difference is illustrated in Fig. 2k, which shows the PDMS and CE membranes after 2 h incubation time with rhodamine B. Using the same imaging setting parameters, the PDMS membrane absorbs more rhodamine B than the CE membrane.

The CE-membrane permeability was assessed by the apical-basolateral transport of two molecules with different molecular weights: FITC–sodium (0.4 kDa) and RITC–dextran (70 kDa). After 4 h of incubation, 25.0% ± 3.5% of the smaller molecules and 12.3% ± 2.9% of the larger molecules were detected in the basolateral chamber (Fig. 2l). The permeability of the membrane was further tested by culturing cells at the apical side of the membrane at the air–liquid interface (see "The CE membrane, a good cell culture support").

The stretchability of the CE membrane was tested by applying a cyclic negative pressure to the basolateral chamber. The membranes of the 40 hexagons deflect simultaneously and homogeneously in three dimensions (Supplementary Movie 1 and Supplementary Fig. 7). Figure 3a shows a numerical simulation of the deflection of the membranes in the array of hexagons. For the 1:1 CE membrane, the applied radial strain reaches 4.9% ± 0.8% for a negative pressure of 1.0 kPa, and almost doubles when −2.0 kPa is applied. The deflection of the CE membrane was compared with the one of a 10-μm-thin PDMS membrane. It appears than the biological substrate deflected more

than the synthetic material (Fig. 3c). At −2 kPa, the radial strain reaches 9.1% ± 2.5% for the CE membrane against only 2.7 ± 0.5% for the PDMS membrane (Fig. 3b). The gold mesh slightly deflected during the experiments, but this did not influence the individual deflection of the membrane in each hexagon (Supplementary Movie 1). When lung alveolar epithelial cells were seeded onto the membrane, −4.0 kPa was needed to induce a 10% linear mechanical strain (Supplementary Fig. 8).

The tunability of the membrane was further investigated by changing the ratio of the proteins concentration. When the elastin or the collagen concentrations were decreased, the membrane became stiffer or softer, respectively, which resulted in smaller or larger deflection, respectively, and thus in smaller or larger linear strains (Fig. 3c and Supplementary Fig. 9). For example, at −1.5 kPa, the deflection of the CE membrane was 31 ± 4.4 μm for a 2:1 ratio, whereas it reached 38.6 ± 3.1 μm for a 1:1 ratio and 51.3 ± 3.9 μm for a ratio 1:3 (Fig. 3c). After being exposed to 10 U mL$^{-1}$ MMP-8, the CE membranes deflected more than an untreated CE membrane. After 45 min of treatment, the membrane deflected at 35.6 ± 3.6 μm (−1 kPa), while the deflection reached 31.0 ± 2.3 μm without treatment (Fig. 3d). By increasing the pressure at −2 kPa, the deflection increased to 63.4 ± 3.0 μm (exposed to MMP-8), against only 42.2 ± 5.8 μm (no treatment). The deflection increased in function of the MMP-8 exposure time (Supplementary Fig. 10). Localised disruption of the membrane could be observed after 1 h MMP-8-exposure (Fig. 2f).

**The CE membrane, a good cell culture support**. Human primary alveolar epithelial cells (hAEpCs) and human lung microvascular endothelial cells (VeraVec) were successfully cultured on

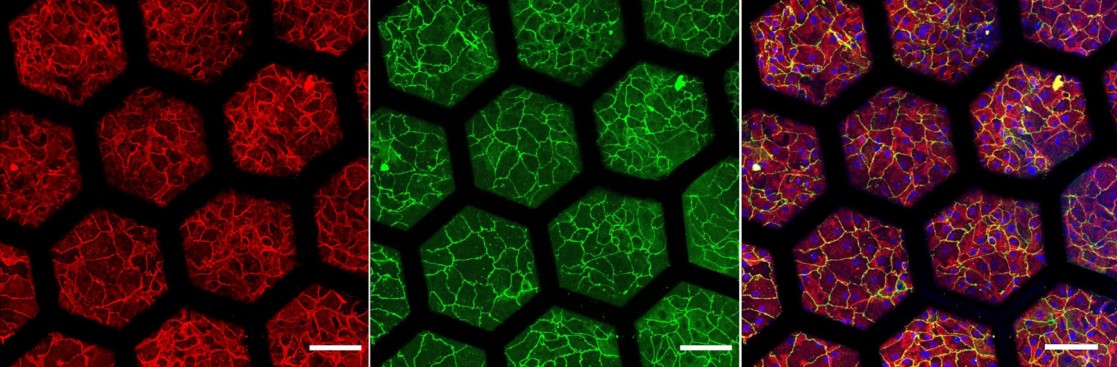

**Fig. 4 Immunostaining of primary human lung alveolar epithelial cells.** hAEpC cultured on the hexagonal mesh with the CE membrane after 4 days and at air–liquid interface for 2 days with expression of adherent junction markers (E-Cadherin, red), tight junctions with zonula occludens-1 (ZO-1, green) and merged (Hoechst, blue; E-Cadherin, red; ZO-1, green). Scale bar: 100 µm.

each side of the membrane. Lung alveolar epithelial cells were also cultured at the air–liquid interface for several days. The cells were confluent and created a functional barrier (Fig. 4 and Supplementary Fig. 11). In these culture conditions, the nutrients diffuse from the basolateral to the apical side of the membrane. On the contrary to PDMS, the CE membrane does not require any preliminary coating for cells to adhere on the membrane. The cells tightly adhered and interacted to the collagen and the elastin as illustrated in the TEM picture of the membrane cross-section and by the observation of phosphorylated focal adhesion kinase (Supplementary Figs. 12 and 13). Lung epithelial cells seeded at high and low density spread and proliferated on the membrane (Fig. 5a and Supplementary Fig. 14). A difference in cellular surface area was observed between day 2 and day 8 between high and low seeding concentration (Supplementary Fig. 14). At high seeding density, cell confluence was reached at day 2. The cellular surface area remained at $1407 \pm 160 \ \mu m^2$, whereas it increased to $2480 \pm 136 \ \mu m^2$ at low seeding concentration. After 2 weeks, primary human lung alveolar epithelial cells were confluent showing nice cell–cell contacts (Fig. 5b, c). Primary human lung alveolar epithelial cells and primary human endothelial cells could both be cultured for at least 3 weeks (Supplementary Fig. 15).

**Reproduction of the lung alveolar barrier.** The typical phenotypes of lung alveolar epithelial cells were investigated using TEM imaging and immunostaining. The characteristic morphologies of type I (ATI) and type II (ATII) lung alveolar epithelial cells—flat and elongated for ATI (Fig. 6b), small and cuboidal for ATII[29] (Fig. 5f)—were recognisable by TEM imaging. Tight junctions, a further characteristic of lung alveolar epithelial cells, were clearly identifiable in Fig. 5d. Zonula occludens (ZO-1) were expressed along the cell borders at day 4 (Figs. 4 and 5e) and day 21 (Supplementary Fig. 15). The barrier formation and the expression of tight junction markers has been checked for four different patients (Supplementary Fig. 16). Surfactant protein-C (SP-C) and lamellar bodies, both typical ATII markers, are shown in Fig. 5e, f.

The permeability of the CE membrane with a monolayer of lung alveolar epithelial cells was further assessed by testing the diffusion capacity of the two molecules used earlier (FITC–sodium, RITC–dextran) (Fig. 5g). The experiment was performed between days 5 and 8 to guarantee the confluence of the epithelial layer. The transport properties of the membrane were significantly affected by the presence or absence of cells (Fig. 5h). For FITC–sodium and RITC–dextran molecules, 25.0% ± 3.5% and 12.3% ± 2.9%, respectively, were transported through

the membrane without cells against 9.4% ± 3.3% and 2.1% ± 1.2%, respectively, with hAEpCs. This result was confirmed with cells from four patients.

To further reproduce the lung alveolar barrier, human lung microvascular endothelial cells were cocultured on the basolateral side of the membrane, with lung alveolar epithelial cells on the apical side. Both cell types reached confluence and populated the whole array (Fig. 6a). Figure 6b illustrates a close-up of the alveolar barrier, with the CE membrane sandwiched between the alveolar epithelium and the microvascular endothelium.

## Discussion

The lung parenchyma comprises of a large number of tiny alveoli organized in a three-dimensional architecture. Thin alveolar walls made of capillary networks and connective tissue separate the alveoli and stabilise the parenchymal construction[11,29]. The breathing movements to which this environment is exposed to, induce a deformation of the alveolar airspaces and of the inter-alveolar septa. The resulting mechanical strain is spatially heterogenous due to the variable thicknesses of the interstitial space and the complex parenchymal architecture[11,30]. This multifaceted and dynamic environment makes the lung alveolar unique and difficult to mimic in vitro. First-generation lung-on-a-chip devices imitate the rhythmic mechanical strain of the alveolar barrier induced by breathing motions[8,9]. Although these systems allow investigation of the mechanobiology of the air–blood barrier for the first time, they are limited by the nature of the PDMS membrane they are made of. The main drawback of PDMS is that it is synthetic which limits its function and the ability to mimic physiological capacities. The ECM of the lung alveolar region has structural and mechanical cell substrate functions but beyond that the ECM is pivotal in determining normal cellular function and differentiation in health and dysregulation in disease[15,31,32]. Another limitation of PDMS membranes is the absorption and adsorption of small molecules and the effect on the ECM as local reservoir of growth factors and bioactive molecules, which are not maintained by the microenvironment at physiological concentrations, and therefore distort effects in the system. This is also a major concern for preclinical drug testing applications, as the effective drug concentration that cells are exposed to is difficult to evaluate[13]. A further drawback is the rather laborious and challenging fabrication process of ultrathin and porous PDMS membranes[9,33]. In addition, first-generation lung-on-a-chip devices imperfectly reproduce the geometric dimensions of the native lung alveoli, as the surface of the culturing membrane creates a unique alveolus of non-physiological dimensions, rather

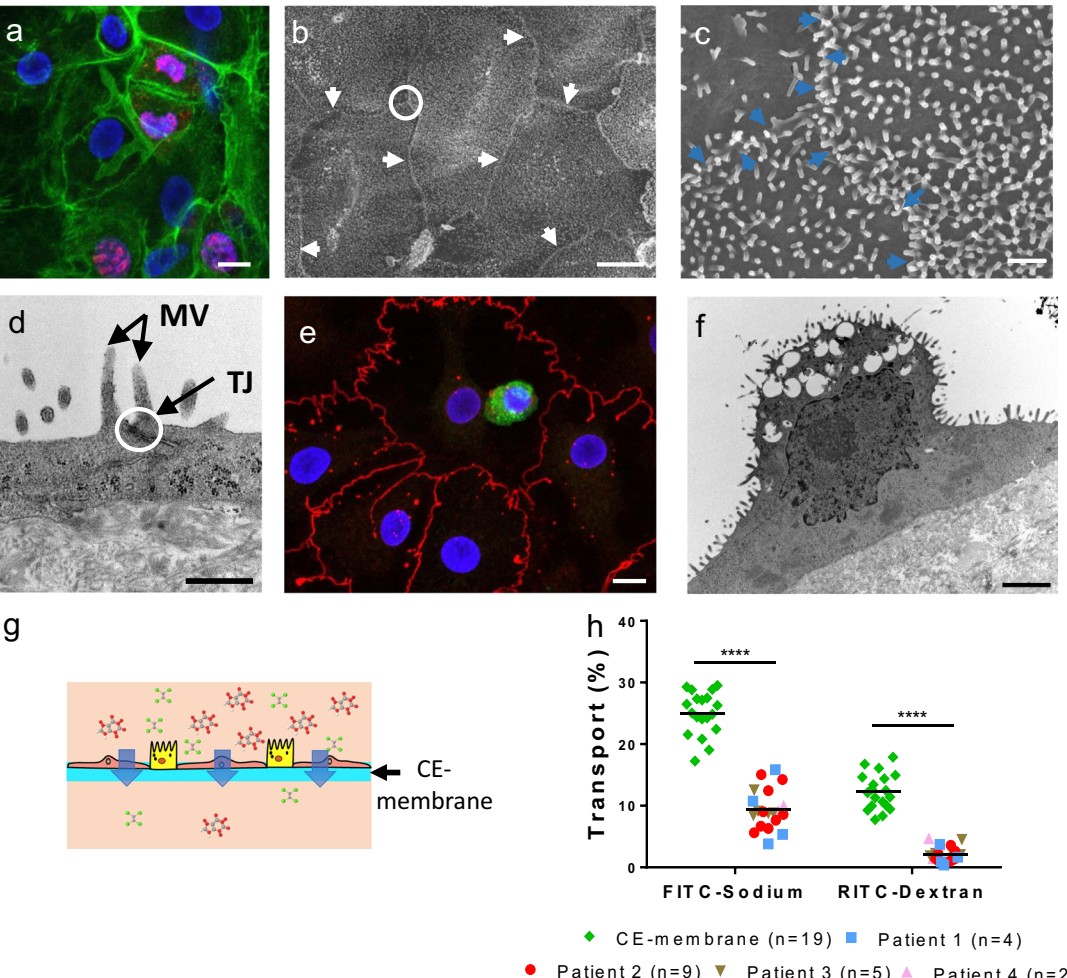

**Fig. 5 Primary lung alveolar epithelial cells. a** Expression of Ki-67 marker on hAEpC at day 4. Actin (green), Ki-67 (red) and Hoechst (blue). Scale bar: 10 µm. **b** SEM picture of hAEpC at day 14, illustrating tight cell–cell contacts. White arrows: cells border; white circle: area zoomed in **c**. Scale bar: 10 µm. **c** Intersection between three cells at day 14, showing their interface and a multitude of microvilli. Blue arrows: cells border. Scale bar: 1 µm. **d** TEM picture of tight junction (TJ) between two hAEpC. Apical microvillis (MV) typical to type II alveolar epithelial cells can clearly be seen. Scale bar: 500 nm. **e** Expression of surfactant protein-C (SP-C, green), tight junction (Z0-1, red) and nuclei (Hoechst, blue) at day 4. Scale bar: 10 µm. **f** TEM picture of a hAEpC type II-like cell at day 4, showing its microvilli and empty spaces, where lamellar bodies were located. Scale bar: 2 µm. **g** Schematic of the transport of molecules across the CE membrane cultured with alveolar epithelial cells. **h** Transport of FITC–sodium and RITC–dextran molecules across the CE membrane ($n = 19$) after 4 h of incubation with hAEpC the experiments were carried with cells from four patients ($n = 20$).

than an array of alveoli of in vivo-like anatomy. This limits investigations of structural and biomechanical changes of alveoli such as those observed in the formation of emphysema[31].

Here, we present a second-generation lung-on-a-chip with an array of alveoli and a stretchable biological membrane that mimics in vivo functionality at an unprecedented level. The CE membrane reproduces the composition and geometrical, biophysical, mechanical and transport properties of the lung alveolar barrier[11]. It recreates the native viscoelastic microenvironment of the cells. Collagen I, the most abundant type of collagen present in connective tissue[34], provides structural stability for the alveoli, and elastin adds elasticity, which is essential for withstanding continuous breathing motions. By tuning the CE ratio and/or adding other ECM molecules, the scaffold stiffness can be tailored to specific applications[35], which is required to model healthy and diseased alveolar environments, such as those present in lung fibrosis[36]. Human primary lung alveolar epithelial cells, which are physiologically more relevant than cell lines[37], are cultured on the biological membrane. Unlike cell lines, these cells present a phenotype that is similar to the original one[38]. The primary healthy lung alveolar epithelial cells used are able to form a

functional barrier, as shown by tight junctions expressed even after several weeks. Type II alveolar epithelial markers, such as lamellar bodies and SP-C, are found after 4 days. TEM pictures reveal remarkable adhesion of the cells to the CE membrane and the reproduction of the epithelial/endothelial barrier. The membrane enables the diffusion of small and larger molecules (FITC–sodium and RITC–dextran) and of epithelial cell nutrients necessary to culture cells at the air–liquid interface, their physiological microenvironment. The results obtained using cells from four patients were similar.

The hexagonal gold mesh with a suspended CE membrane provides cells with small alcoves containing an environment similar to that found in an alveolus as measured by a number of different parameters. First, the size of each small alcove is the same order of magnitude as the diameter of lung alveoli, reported to be around 100–200 µm[30,39,40]. Second, the borders of the alcoves mimic the alveolar walls[41,42] that separate alveoli from each other and strengthen the stability of the structure. Third, the three-dimensional mechanical stress created within each alcove is distributed in a physiological strain gradient. Like in vivo, the mechanical strain the cells are exposed to on the membrane varies

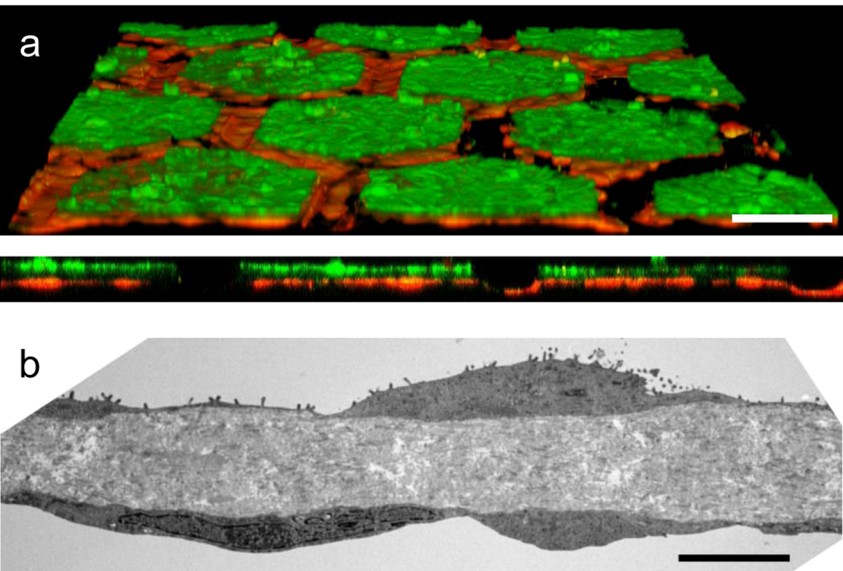

**Fig. 6 Air–blood barrier reproduction. a** Confocal pictures (perspective view and cross-section) of a coculture of hAEpC (E-Cadherin in green) with human primary endothelial cells (Rfp-label in red) on the hexagonal mesh with the CE membrane. Scale bar: 100 μm. **b** TEM picture of hAEpC type I-like cells in coculture with human lung endothelial cells at day 4. Scale bar: 5 μm.

spatially, with a mechanical strain reaching 10%, which is in the range of the physiological strain[11,30]. This environment, combined with the CE membrane, gives the cells more physiological cell culture conditions and may also enable the recreation of biological events that at their onset only involve a limited number of cells. Investigations of phenotypic changes underlying lung cell pathologies and their effect on downstream signalling cascades become possible in tissue-specific primary cell culture microenvironments. Another key feature of the CE membrane is its biodegradability. In the lung, the degradation of the ECM is an important event in tissue remodelling[31]. As a proof of concept, MMP-8, a known neutrophils collagenase, was chosen to demonstrate the degradation of the CE membrane. The biodegradability of the membrane is of high interest for the investigation of the alveolar barrier remodelling, that typically takes place in a number of lung diseases, such as emphysema, lung fibrosis and lung cancer[32,43–45]. The biochemical and mechanical properties of the ECM are tightly coupled to disease progression. We anticipate, that the biodegradability property of the membrane combined with the ease to tune its mechanical properties will enable to recapitulate the healthy and diseased cellular microenvironments with greater accuracy than what is achievable to date.

The simple and reproducible production process of the CE membrane makes it an easy-to-use tool for academic laboratories as well as for larger scale applications, like screening. In fact, the creation of the biological membrane is based on a bottom-up approach (surface tension), rather than the top-down approach (photolithography) used to produce thin, porous PDMS membranes[33]. This elegant fabrication technique is much less cumbersome than that used to produce polymeric membranes. The dehydrated ECM array is robust and can be stored for several weeks at room temperature. These unique features are of high relevance for the lung bioengineering and the OOC research communities. In addition, the CE membrane has great versatility as thickness can easily be tuned by adapting the volume of the CE solution pipetted onto the gold mesh to suit any number of experimental requirements. The thinnest membrane obtained has a thickness comparable to the thinnest porous PDMS membrane used on lung-on-a-chip reported thus far[10]. Unlike synthetic polymers, such as PDMS, the CE membrane does not require any preliminary coating prior to cell seeding. Moreover, the absorption and adsorption issue observed with the PDMS membranes is almost absent. Taken together, these characteristics make the CE membrane a credible alternative to PDMS, with advantages of usability, production and stability.

We have developed a lung alveoli array that displays characteristics of the lung parenchyma with analogy to native alveolar tissue in a number of physiological parameters. Three key features of production and properties were considered in the development of this new generation of organ-on-a-chip. First, a suspended culturing membrane was created by surface tension force. Second, the CE membrane mimics the native, deformable ECM of the lung parenchyma. Third, an array of alveoli with more physiological geometric proportions was created by the gold mesh. The replacement of PDMS membranes is desirable in in vitro barrier models, and this CE membrane is a versatile and generic solution that can be expanded to mimic other in vivo barriers. This technology has the potential to become a powerful tool to investigate basic science questions, screen compounds in drug development, model lung diseases and identify the best treatment option for each patient in precision medicine.

## Methods

**Production of the CE membrane**. The CE membrane was produced as follows (Fig. 1). The membrane was based on rat-tail collagen type I, high concentration (Corning, New-York, NY, USA), and bovine neck elastin powder/lyophilised (Sigma-Aldrich, Buchs, Switzerland). The two molecules were mixed at a final concentration of 3.5 mg mL$^{-1}$ in a pH 7.4 buffer. A 18-μm-thin gold mesh (Plano GmbH, Wetzlar, Germany) with hexagonal pores of 225 μm (inner diameter) and 260 μm (outer diameter) was used as a scaffolding to create the biological membrane. The gold mesh was successively treated with 5% 3-aminopropyl triethoxysilane (APTES) (Sigma-Aldrich) and 0.1% glutaraldehyde (Sigma-Aldrich) to ensure attachment of the membrane. The CE solution was pipetted directly on top of the gold mesh. Its thickness was tuned by adapting the volume of the CE solution pipetted. After pipetting, the chip was immediately placed at 37 °C, 100% humidity and 5% CO$_2$ for 2 h 15 min to allow gelation of the membrane. Then, the membrane was placed for 48 h at room temperature to dry. Membranes were stored at room temperature. Before use, membranes were rehydrated with cell culture media for 2 h at 37 °C.

**Micro-device fabrication**. To create the air–blood barrier on-a-chip, a PDMS (PDMS Silgard 184, Dow Corning, Midland, MI, USA) plate was attached to a

polycarbonate bottom with double tape (Arcare 90445-5, Adhesives Research, Glen Mark, PA, USA). The gold mesh with the CE membrane was sandwiched between the two chambers (Supplementary Fig. 2). This design enabled the compartmentalisation of the culture medium in the apical and the basolateral chamber. The top layer was produced by PDMS soft lithography. Briefly, a prepolymer was mixed with a curing agent at a weight ratio of 10:1 and placed in a vacuum chamber to remove air bubbles. After degassing, the PDMS mixture was cast in a mould with two dowel pins located at the border of the chip as alignment features. After an incubation at 60 °C overnight, the PDMS mixture was fully cured, and cut into a rectangular shape of $20 \times 15 \times 3.2$ mm. The bottom layer was made of polycarbonate with a central hole of 2 mm and two 1.5 mm additional holes on both sides of the lower part to allow access to the membrane. The top layer can easily be detached from the bottom layer to reduce the focal distance required for confocal imaging. Prior to being used, the chip was sterilised by autoclaving (CoolCLAVE, Genlantis, San Diego, CA, USA).

**Cell culture**. Primary hAEpCs were isolated from patient tissue according to a protocol reported previously[10,46]. Briefly, alveolar epithelial type II (ATII) cells were isolated from tissue obtained from healthy areas removed from patients undergoing lung tumour resection surgery. All patients gave informed written consent for usage of surgical material for research purposes, which was approved by ethical committee from the Ärztekammer des Saarlandes. All procedures were carried out in accordance with institutional guidelines from Saarland (Germany) and from the Canton of Bern (Switzerland). Cells were cultured in Small Airway Growth Medium (SAGM™, Lonza, Basel, Switzerland) with BulletKit (CC-3118, Lonza), supplemented with 1% FBS (Sigma) and 1% P/S. RFP-labelled human lung microvascular endothelial cells (VeraVec, Angiocrine Biosciences Inc., San Diego, CA, USA) were cultured in EGM2 medium (Lonza) supplemented with growth factors according to the manufacturer's instructions (EGM2-MV BulletKit, Lonza). All cell manipulations were performed in a sterile flow hood, and cells were maintained at 37 °C, 100% humidity and 5% $CO_2$.

For monoculture, hAEpCs were seeded with a density of 270,000 cells $cm^{-2}$ or at 100,000 cells $cm^{-2}$ (low concentration condition). The cells were incubated for 24 h, allowing them to adhere to the membrane, and reached confluence after 48 h. To create a coculture, the chip was flipped, and VeraVec cells were seeded on the basal side of the CE membrane at $1.0e^6$ cells $mL^{-1}$. After 24 h, the chip was flipped again, and the medium was changed to remove all non-attached cells. After 48 h, epithelial cells were seeded on the apical side at 270,000 cells $cm^{-2}$. After 24 h, 50/50 medium (half EGM2-MV supplemented and half SAGM supplemented) was used in both monoculture and coculture. Medium was changed daily. Primary human lung alveolar epithelial cells were cultured on a PDMS membrane using a protocol described earlier[10].

**Measurement of the membrane thickness**. The thickness of the membrane was measured with reflective light microscopy. Briefly, the membrane was cut at its centre and imaged using the Axioplan microscope (Zeiss, Oberkochen, Germany). For each membrane, several points were measured using Axiovision software. Confocal imaging (z-stack) with LSM710 (Zeiss) was used to confirm the thickness of the membrane. Images were analysed with ImageJ software.

**Transparency**. The optical transparency of the membrane was evaluated using a light spectrometer (M1000 Infinite, Tecan) in the range of 350–700 nm. Membranes were produced by pipetting a solution of the specific material to be tested on the bottom of a 96-well plate. The volume of the solution was adapted to obtain a 10-μm-thin membrane.

**Absorption/adsorption**. The ab- and adsorption of small molecules by the membranes was quantified by immerging them in 10 μM rhodamine B (Sigma-Aldrich) in PBS for 2 h at 37 °C. A CE membrane with ratios 1:1 and 2:1; a collagen membrane; a polyester membrane with 0.4 and 3 μm pore sizes and 40, 10 and 3.5 μm porous PDMS membranes were tested. After immersion in rhodamine B, membranes were washed twice in PBS for 5 min. The fluorescence of each membrane was measured using a standard spectrometer (Infinite M200, TECAN, Mannedorf, Switzerland) with an excitation wavelength of 544 nm and an emission of 576 nm. Pictures of the membranes after immersion were taken with a Leica DMI400 (Leica Microsystems, Buffalo Grove, IL, USA). The PDMS membranes were fabricated by spinning PDMS attached to a silicon wafer at 1650 rpm (resp. 6700 rpm) for 60 s to obtain a 40 μm (resp. 10 μm) membrane. The membrane was then cured for 24 h at 60 °C. The 3.5 μm porous membrane was produced according to a procedure reported previously[9].

**Biodegradability assay**. CE membranes were produced as described above. After reswelling the membrane overnight, the membranes were incubated at 37 °C with a solution of MMP-8 (Sigma-Aldrich, initial concentration 280 U $mg^{-1}$) diluted in PBS+ in three different concentrations: 50, 10 and 5 U $mL^{-1}$. Four membranes were used per condition. Membranes immerged with PBS+ and wells without membranes were used as controls. The absorbance of the membranes was analysed with a light spectrometer (M1000 Infinite, Tecan) at 550 nm.

**Permeability assay**. Once the cells were confluent the lower chamber was filled with cell culture medium. The upper chamber was filled with 1 μg $mL^{-1}$ FITC–sodium (0.4 kDa, Sigma-Aldrich) and 1.5 mg $mL^{-1}$ RITC–dextran (70 kDa, Sigma-Aldrich) in 50/50 medium (half EGM2-MV supplemented and half SAGM supplemented). The device was incubated for 4 h, after which the solution in the upper channel was removed and the top chamber was washed three times with PBS. Subsequently, the solution from the lower chamber was collected. The samples were tested for fluorescence with a multi-well plate reader (M1000 Infinite, Tecan). The FITC–sodium and RITC–dextran were excited at 460 and 553 nm, respectively. Emission was measured at 515 and 627 nm, respectively. The permeability of the air–blood barrier was expressed in terms of relative transport, in that the amplitude of the fluorescent signal of the basal chamber solution was normalised to the fluorescence intensity signal of the initial solution of the apical chamber. The time point of the permeability assays performed on the CE membrane was defined based on transepithelial electrical resistance (TEER) values obtained in hAEpC cultured in parallel on inserts. The formation of a confluent and tight epithelial layer results in a steady increase of the TEER value until reaching a plateau indicating a functional barrier. All patient's cells used in this study were tested with this method.

**Numerical simulation**. A stationary numerical simulation, using COMSOL Multiphysics 5.3 (COMSOL Multiphysics GmbH, Switzerland), was performed to visualize and illustrate the deformation of the CE membrane during breathing.

**Measurement of deflection**. The membrane was cyclically deflected using a homemade electro-pneumatic system generating a cyclic negative pressure that can be tuned from 1 to 30 kPa. The deflection measurement was performed by the evaluation of the height difference between stretched and unstretched membrane. Pressure was applied for 20 s, followed by a resting time of 1 min. For each membrane, a minimum of three hexagons located at the centre of the membrane were measured (except for the CE membrane (1:3) where only one hexagon in the centre was measured). On each hexagon, two points were measured: one at the centre of the membrane and one on the gold mesh hexagon. These values were obtained with an AxioPlan2 Zeiss microscope. Linear stress was calculated based on the absolute deflection of the membrane, which was approximated as a circular segment (Supplementary Fig. 17).

**Immunofluorescence**. All immunostaining steps were conducted at room temperature. The chips were washed three times with PBS, fixed with 4% paraformaldehyde (Sigma-Aldrich) for 10 min and rinsed again three times with PBS. The cells were permeabilised with 0.1% Triton X-100 (Sigma-Aldrich) for 10 min and washed three times with PBS. After 45 min of blocking in a 2% BSA (Sigma-Aldrich) solution, primary antibodies were diluted in the blocking solution. The chip was incubated for 1.5 h. Following incubation, devices were washed three times with PBS, then incubated for 1 h with the associated secondary antibody. A 1:2000 dilution of Hoechst was added to image cell nuclei. Finally, the chip was washed with PBS. The top layer was detached from the bottom to image the cells on the membrane. Images were obtained using a confocal microscope (CLSM, Zeiss LSM710).

**Scanning electron microscope**. For SEM acquisition, samples were fixed with 2.5% glutaraldehyde (Merck) in 0.1 M cacodylate buffer (Merck) at pH 7.4 for 1 h at room temperature. After rinsing three times in a 0.1 M cacodylate buffer, the samples were post-fixed for 10 min in a 1% osmium tetroxide solution in 0.1 M sodium cacodylate buffer. After rinsing three times with Acqua Dest (Medical Corner 24, Oer-Erkenschwick, Germany), the chips were dehydrated at room temperature in 50, 70, 80 and 95% ethanol for 10 min each. Next, they were immersed in 100% ethanol three times for 10 min. Finally, the samples were immersed in hexamethyldisilane for 10 min and then dried at room temperature. Samples were mounted onto stubs with adhesive carbon (Portmann Instruments, Biel-Benken, Germany) and coated by electron beam evaporation with platinum/carbon (thickness of coating: 26 nm). Pictures were taken with the DSM982 Gemini digital field emission scanning electron microscope (Zeiss) at an acceleration of 5 kV and a working distance of 3 mm.

**Transmission electron microscopy**. For TEM acquisition, the chips were fixed with 2.5% glutaraldehyde (Agar Scientific, Essex, UK) in 0.15 M HEPES (Sigma-Aldrich) buffer (670 mOsm, pH 7.35). The samples were placed at 4 °C. Samples were post-fixed for 1 h in a 1% osmium tetroxide solution in 0.1 M sodium cacodylate buffer (Merck) and rinsed three times in the same buffer. Next, the chips were dehydrated at room temperature with an ethanol concentration series (70, 80 and 96%) for 15 min each. Then, they were immersed in 100% ethanol (Merck) three times for 10 min. The chips were embedded in an epoxy solution and incubated at 60 °C for 4 days. For samples without cells, the chips were directly embedded in the epoxy solution. After removing the PDMS surrounding the gold mesh, ultrathin sections (70 nm) were cut with an ultramicrotome UC6 (Leica Microsystems) and mounted on 1 mm single-slot copper grids. Pictures were taken with a Philips EM 400 transmission electron microscope.

**Statistics and reproducibility**. Data are presented as mean ± standard deviation (SD). The error bars represent the SD. Two-tailed unpaired Student's $t$ test was used to assess the significance of differences. Statistical significance was defined as follows: $*p < 0.05$, $**p < 0.01$, $***p < 0.001$ and $****p < 0.0001$. Statistical analysis was performed using GraphPad Prism 6 software. The sample sizes and numbers are indicated in detail in each figure legend.

**Reporting summary**. Further information on research design is available in the Nature Research Reporting Summary linked to this article.

## Data availability

The source data underlying figures are provided as Supplementary Data 1. Any remaining information is available from the corresponding author upon reasonable request.

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

## Acknowledgements

The Swiss National Science Foundation is acknowledged for its financial support (project number: 185365). The authors thank Prof. em. Dr. phil. nat. Peter Gehr from the Institute of Anatomy of the University of Bern for graciously providing the SEM picture of the lung alveoli (Fig. 1b). They also acknowledge Beat Hänni from the same institute for his help processing the samples and taking the TEM pictures (Figs. 5d, f and 6b and Supplementary Fig. 12). The authors thank Prof. Dr. med. vet. Michael Stoffel and Helga Mogel from the Division of Veterinary Anatomy of the University of Bern for their help in the preparation and acquisition of the SEM data (Figs. 2i and 5b, c). They also acknowledge the Microscopy Imaging Center (MIC) at the University of Bern. The authors thank Jan Schulte for his help in cells surface measurement. The authors thank Dr. Usha Sarma for helping revising the manuscript and to Dr. Anne Morbach for the illustration of Fig.1c–g.

## Author contributions

O.T.G. had the original idea of using a hexagonal mesh to mimic the lung alveoli. J.D.S., P.Z., N.H. and O.T.G. designed the experiments. S.A. carried out the proof of concept experiments, and P.Z. and S.W. set up the device. P.Z. carried out the experiments and collected the data, except Fig. 3b, c obtained by S.W. and G.T. Authors P.Z., J.D.S., N.H. and O.T.G. analysed and interpreted the data. J.D.S. modelled numerically the deflection of the stretchable membrane. P.Z. and O.T.G. wrote the manuscript. N.S.-D., C.-M.L. and H.H. provided the study material. T.G. and R.A.S. revised the manuscript. All authors reviewed the manuscript.

## Competing interests

O.T.G. and J.D.S. are co-authors of a patent that describes the use of the mesh as in vitro barrier and whose rights are with the University of Bern and AlveoliX AG. All other authors declare no competing interests.
