## [Peer Review File · Communications Biology]

REVIEWERS' COMMENTS:

This manuscript describes a lung-on-a-chip array, which can simulate the air-blood barrier by using a collagen-elastin membrane. Generally, the membrane plays a key role in the lung-on-a-chip. The CE membrane used by the authors seems to have good biocompatibility and deformability. In addition, the authors also used primary cells from patients to build the air-blood barrier to verify the effectiveness of the established system. Some minor revisions are expected before the manuscript is accepted.

1. Why should the author emphasize the second-generation chip? Although the membrane material has been changed, I personally think that the model of the lung-on-a-chip has not undergone a fundamental change. Can the author consider changing the title and some description?
2. MMPs also play an important role in the development of tumors. CE membrane can be degraded by MMP-8. Will it limit the application of the established microchip in lung tumor research? Can the author increase some discussion?
3. The author just uses cells from patients to construct a blood-air barrier. Are there more applications, such as drug experiments to guide clinical treatment?

Answers to the reviewers

Manuscript ID: COMMSBIO-20-2667-T

We would like to thank the reviewer for his/her comments and the time invested to review our manuscript. In the following we answer (in blue) all the questions/comments raised by the reviewer. The modified text is highlighted in red in the main document.

Answer to Reviewer

1. Why should the author emphasize the second-generation chip? Although the membrane material has been changed, I personally think that the model of the lung-on-a-chip has not undergone a fundamental change. Can the author consider changing the title and some description?

Thank you for your comment. Yes, we are happy to clarify why we call this system the second-generation lung-on-chip (LOC). You are absolutely correct, the sole modification of the material of the membrane would indeed only be an incremental change of the system not worthy of representing a new generation of LOC. In sharp contrast, the system reported here, mimics central aspects of the air-blood barrier that have never been reported before: 1) an array of alveoli, with dimensions similar to those found *in vivo*, a membrane 2) only made of proteins of the lung ECM, among others elastin, that enable the membrane to be 3) stretchable and 4) biodegradable. These unique features are seen as being of high relevance by the lung bioengineering community. They will enable mimicking the air-blood barrier remodeling in health and disease and be “a promising tool to evaluate interactions between the lung ECM and cells” (see Wagner DE, Ikonomou L, Gilpin SE, et al. Stem Cells, Cell Therapies, and Bioengineering in Lung Biology and Disease 2019.ERJ Open Res2020; 6: 00123-2020 [<https://doi.org/10.1183/23120541.00123-2020>])

Furthermore, the engineering innovation is of significance importance as well. The elegant fabrication of the membrane using a bottom-up approach (surface tension) rather than a top-down approach (photolithography), makes its fabrication much less cumbersome than that of polymeric (PDMS) stretchable membranes. This fabrication process is versatile enough to enable the creation of different types of membranes, with different compositions, stiffness, thickness, etc. Finally, the CE membrane does not suffer the problematic ad-/absorption of small molecules like their PDMS counterparts.

All these reasons make from our perspective this technology worthy of being named “second-generation LOC”. To clarify this, we have modified the title and several sections of the manuscript.

Modifications:

The title was modified as follow: “**Second-generation lung-on-chip with an array of stretchable alveoli made with a biological membrane**”

The introduction was modified as follow (text added in the second paragraph):

“Although these systems represent a crucial advance in cell culture research, they are still far from mimicking the whole *in vivo* intricacy. An important *in vivo* feature that is not reproduced in air-blood barrier models is the array of tiny alveoli. Indeed, reported lung-on-a chips from the first-generation^{8,9} simulate a single alveolus with an epithelium area much larger than that of an alveolus *in vivo*. Given the close relationship between lung microstructure, mechanical forces, alveolar epithelial phenotype and lung functions^{11,12}, the emulation of the alveolar network would be of great benefit. A further limitation of those systems is the use of an artificial basal membrane made of a thin, porous and stretchable polydimethylsiloxane (PDMS) film.”

(text added in the fourth paragraph):

“Here, we report about a lung-on-a chip of the second-generation that mimics the following central aspects of the air-blood barrier: 1) an array of alveoli, with dimensions similar to those found in-vivo, 2) a biological membrane made of proteins of the lung extracellular matrix, collagen and elastin, enabling the membrane to be 3) biodegradable and 4) stretchable. The creation of the biological membrane is based on a bottom-up approach fabrication technique.”

The discussion is modified as follow (fourth paragraph):

“In fact, the creation of the biological membrane is based on a bottom-up approach (surface tension), rather than the top-down approach (photolithography) used to produce thin, porous PDMS membranes³³. This elegant fabrication technique is much less cumbersome than that used to produce polymeric membranes. The dehydrated extracellular matrix array is robust and can be stored for several weeks at room temperature. These unique features are of high relevance for the lung bioengineering and the organs-on-chip research communities.”

2. MMPs also play an important role in the development of tumors. CE membrane can be degraded by MMP-8. Will it limit the application of the established microchip in lung tumor research? Can the author increase some discussion?

Thank you for this interesting remark. This may indeed be a limitation of a collagen-elastin membrane. However, the membrane can easily be modified, either by decreasing the collagen concentration and/or by adding other ECM components, such as laminin, fibronectin, fibrin,..., which would prevent collagenase-biodegradation. In the present case, MMP-8 was used as proof of concept to show the biodegradability of the membrane. This property opens the doors to investigate barrier remodeling and ECM-cell interaction. The ECM is a highly dynamic structure constantly remodeled to control tissue homeostasis. Any dysregulation could exacerbate disease progression. Another possibility that might be desired as well, is that the degraded ECM could be replaced by proteins secrete by cells. Additionally, the loss of the ECM will change the mechanical properties of the barrier to recreate a unique *in vivo* like microenvironment in term of composition but also mechanical properties.

Modifications: The following text and reference were added in the Discussion section and a reference related to MMP8 in cancer added (Bonnans, C., Chou, J. & Werb, Z. Remodelling the extracellular matrix in development and disease. *Nat. Rev. Mol. Cell Biol.* **15**, 786–801 (2014)):

“The biodegradability of the membrane is of high interest for the investigation of the alveolar barrier remodeling, that typically takes place in a number of lung diseases, such as emphysema, lung fibrosis and lung cancer.^{31,43,44,45} The biochemical and the mechanical properties of the ECM are tightly coupled to disease progression. We anticipate, that this biodegradability property combined with the ease to tune the mechanical properties of the membrane will enable to recapitulate the healthy and diseased cellular microenvironments with greater accuracy than what is achievable to date”.

3. The author just uses cells from patients to construct a blood-air barrier. Are there more applications, such as drug experiments to guide clinical treatment?

Thank you for this important comment. In this study we highlight the development of a healthy air-blood barrier with a new biological membrane. However, you are perfectly right; in the future, this model may be used for a range of applications, starting with basic science questions, drug development screening and precision medicine applications. In the latter case, the model could be used to identify the best possible treatment for each patient, using patients cells (either obtained via biopsy, or using iPSc-derived cells). The used of primary cells offer a number of advantages compared to cell line including a phenotype that is close to the original tissue, the ability to differentiate to an *in vivo* like tissue and an increased donor diversity reflecting the natural diversity of human population (Eglen et al, 2011; Nawroth et al, 2019). Then, primary cells offer the possibility to study the impact of several factors such as the age, medical history, smoking habits, race, sex... on drug response.

Modifications: The following text and references were added in the Discussion section:

“Human primary lung alveolar epithelial cells, which are physiological more relevant than cell lines³⁷, are cultured on the biological membrane. Unlike cell lines, these cells present a phenotype that is similar to the original one.³⁸”

In the conclusion:

“The replacement of PDMS membranes is desirable in *in vitro* barrier models, and this CE-membrane is a versatile and generic solution that can be expanded to mimic other *in vivo* barriers. This technology has the potential to become a powerful tool to investigate basic science questions, screen compounds in drug development, model lung diseases and identify the best treatment option for each patient in precision medicine.”